# An OpenCV-Based Approach for Automated Cardiac Rhythm Measurement in Zebrafish from Video Datasets

**DOI:** 10.3390/biom11101476

**Published:** 2021-10-07

**Authors:** Ali Farhan, Kevin Adi Kurnia, Ferry Saputra, Kelvin H.-C. Chen, Jong-Chin Huang, Marri Jmelou M. Roldan, Yu-Heng Lai, Chung-Der Hsiao

**Affiliations:** 1Department of Chemistry, Chung Yuan Christian University, Chung-Li 320314, Taiwan; smalifarhan@gmail.com (A.F.); ferrysaputratj@gmail.com (F.S.); 2Department of Bioscience Technology, Chung Yuan Christian University, Chung-Li 320314, Taiwan; kevinadik-adi@hotmail.com; 3Department of Applied Chemistry, National Pingtung University, Pingtung 900391, Taiwan; kelvin@mail.nptu.edu.tw (K.H.-C.C.); hjc@mail.nptu.edu.tw (J.-C.H.); 4Faculty of Pharmacy, The Graduate School, University of Santo Tomas, Espana Blvd., Manila 1015, Philippines; mmroldan@ust.edu.ph; 5Department of Chemistry, Chinese Culture University, Taipei 11114, Taiwan; 6Center for Nanotechnology, Chung Yuan Christian University, Chung-Li 320314, Taiwan; 7Research Center for Aquatic Toxicology and Pharmacology, Chung Yuan Christian University, Chung-Li 320314, Taiwan

**Keywords:** zebrafish, cardiac rhythm, arrhythmia, OpenCV, computer vision, Daphnia

## Abstract

Cardiac arrhythmia has been defined as one of the abnormal heart rhythm symptoms, which is a common problem dealt with by cardiologists. Zebrafish were established as a powerful animal model with a transparent body that enables optical observation to analyze cardiac morphology and cardiac rhythm regularity. Currently, research has observed heart-related parameters in zebrafish, which used different approaches, such as starting from the use of fluorescent transgenic zebrafish, different software, and different observation methods. In this study, we developed an innovative approach by using the OpenCV library to measure zebrafish larvae heart rate and rhythm. The program is designed in Python, with the feature of multiprocessing for simultaneous region-of-interest (ROI) detection, covering both the atrium and ventricle regions in the video, and was designed to be simple and user-friendly, having utility even for users who are unfamiliar with Python. Results were validated with our previously published method using ImageJ, which observes pixel changes. In summary, the results showed good consistency in heart rate-related parameters. In addition, the established method in this study also can be widely applied to other invertebrates (like Daphnia) for cardiac rhythm measurement.

## 1. Introduction

Heart rate is a measurement of cardiac activity represented in the number of beats per minute (bpm) [1]. Generally, heart rate has some subtle differences between beats, known as heart rate variability (HRV), which shows a specific range for the average heart rate within the species. For example, the range in humans is 60–100 bpm, and if the heart rate is beyond this range, it is considered bradycardia or tachycardia. Bradycardia is a condition when the heart rate is lower than the lowest threshold (<60 bpm) of HRV. On the other hand, tachycardia is when the heart rate is higher than the highest threshold (>100 bpm). In addition, more severe cases are reported as arrhythmia, showing high HRV, and all these conditions have life-threatening risks related to symptoms causing cardiovascular-related death. Therefore, it can be treated as a warning for cardiac electrophysiologists and cardiologists [2,3,4].

The factors that affect heart rate measurement are based on biological conditions, such as physical activity, stress, sleep, heart diseases, illness, and therapeutic applications. Photoplethysmography (PPG), a non-invasive optical method, is primarily applied to record the volumetric behavior in peripheral blood circulation [5]. The PPG method was based on the reflection of a specific wavelength in segments of the human body, and it can be used to measure the blood volume pulse (BVP) during the cardiac cycle [6]. PPG can be used with low-intensity infrared (IR) light for heart rate measurement, and the wave pattern reflected from tissue during the cardiac cycle has similarity to electrocardiogram (ECG) peaks which could be used to diagnose cardiac arrhythmia. Verkruysse et al. [7] explained that plethysmographic signals could also be recognized in video with the help of the conventional color camera, revealing new heart rate measuring probabilities. Recent advances of PPG made it applicable in biomedical research to predict early vital signs of arrhythmia in patients [8,9]. The non-invasive approach used to estimate human heart rate was created by Poh et al. [10] by tracking user face videos with a blind source distinction of the color channels into self-supporting elements. Wu et al. [11] worked with their Eulerian video magnification (EVM) framework to observe the human pulse and amplify the color signals that are prominent for the view of minor changes. Furthermore, Balakrishnan et al. [12] used a unique method for extracting the heartbeat from different videos by putting a mark on the head of a human and then measure its detailed movement. Chen et al. [13] proposed an illumination-invariant approach for measuring the heart rate evaluation. It was comprised of both the near-infrared channel and the detailed data obtained by a RealSense 3D (RGBD) camera. The feasibility of measuring human heart rate with the help of video is now advanced, which allows measuring heartbeat with low light intensity. 

Moreover, Davila et al. [14,15] launched a non-contact method of estimating the arterial pulse that gave a figure of HRV parameters. However, developing non-contact sensors comprising the physiology of the arterial pulse and the regular changes in light absorption and reflection in the dermis is still challenging. Moreover, the advanced techniques of detecting the heartbeat are implemented by independent component analysis (ICA) on the color channels of recorded videos to get the PPG signal are commonly practiced in various biomedical research fields. Alghoul et al. [16] proposed and compared the EVM-based method to the ICA-based approaches to enhance heart rate detection and HRV analysis in humans. 

Zebrafish have proven to be a good vertebrate model for cardiovascular research, which has a transparent body in the larva stage, enabling visual observation of cardiac activity [17,18,19,20]. The heart rate of zebrafish is closer to humans (120–180 bpm) compared to mice (300–600 bpm), which are one of the most commonly used animal models [21]. Most methods calculate heart rate data by recording region-of-interest (ROI), followed by digital image processing to obtain cardiac-related data [22,23,24,25]. In our previous works, two ImageJ-based methods in terms of time series analyzer (TSA) [26] and kymograph [21] were reported to conduct cardiac rhythm measurement in zebrafish. However, some limitations in image-based methods on measuring cardiac rhythms were encountered, such as the data variation from different ROI locations and tedious manual or semi-automated operation processes [27,28,29,30]. Recently, advancements in Artificial Intelligence (AI) and computer vision library [31] provide wide application in the healthcare environment to reduce cost and time for more efficient clinical practices. Furthermore, there are several available spatial measurement algorithms used in computer vision, such as Channel and Spatial Reliability Tracking (CSRT) [32], Joint Approximate Diagonalization of Eigen-matrices (JADE) [33], and Cross-Correlation algorithm [34]. ICA is one of the promising methods to determine heart rate from facial videos and non-contact devices [35]. In addition, OpenCV is a very powerful library for the analysis of image or video datasets in AI and deep learning. It has more than 2500 available algorithms that provide users with a wide range of applications depending on their necessities [36]. It is supported by Windows, Linux, and Mac-OS, and other operating systems and interfaces for C++ Python, Java, and MATLAB [37]. Inspired from previous computer vision tools developed for cardiac rhythm detection in humans, in this study, OpenCV is used to detect zebrafish cardiac activity from a video dataset in Python programming language [38] for the first time. The feature of simultaneously selecting ROIs for atrium and ventricle in video with presenting automatic results are the hallmarks of this study. Additionally, user-friendly tools and less tedious methods will benefit researchers to analyze huge datasets.

## 2. Materials and Methods

### 2.1. Zebrafish Embryo Maintenance

All experiments involving zebrafish were performed according to the guidelines approved by the Institutional Animal Care and Use Committees (IACUCs) of Chung Yuan Christian University (Approval No. 109001, issue date 15 January 2020). Wild-type AB strain zebrafish were maintained in a continuously aerated and filtered water system. According to previously reported protocols, the water temperature was maintained at 26 ± 1 °C with 14/10 h of light/dark cycle [39]. Male and female zebrafish with a ratio of 2:1 and placed into the breeding chamber at nighttime before breeding. The following day, the separator was removed, and the embryos were collected after 2 h. After harvesting, embryos were immediately cleaned from the impurities with double distilled water and put into an incubator with a temperature of 28 °C until the time of treatment. To ease the time calculation, all embryos were counted as 0 h post-fertilization (hpf) at the time of fertilization. 

### 2.2. High-Speed Videography

During recording the heart chamber of zebrafish, 3% methylcellulose was used as a mounting solution, and larvae at 72 hpf were set on a petri dish with the lateral view facing up. A high-speed charged coupled device (CCD; AZ Instrument, Taichung City, Taiwan) camera mounted on the inverted microscope (ICX41, Sunny Optical Technology, Zhejiang, China) was used to record the zebrafish heart chamber. The LPlan modulation objective lens with 10× magnification was used to get the image of the zebrafish larvae cardiac chamber. The recording was done using HiBestViewer Software (AZ Instrument, Taichung City, Taiwan) at 200 frames per second (fps) for 10 s.

### 2.3. Calculation of Heart Rhythm Using ImageJ

For data validation, the heart rhythm was calculated using Time Series Analyzer Plugin on FIJI distribution of ImageJ software (https://imagej.nih.gov/ij/plugins/time-series.html, accessed on 30 July 2021) by calculating the dynamic change of the brightness intensity. The calculation of the heart rate followed our previously published method [26]. The timing of each beat was obtained using the BAR Plugin on ImageJ (available online: https://imagej.net/BAR, accessed on 30 July 2021) and the Poincare Plot plugin from OriginLab software (Originlab Corporation, Northampton, MA, USA) was used to assess the heart rate variability. Furthermore, the atrium-ventricle relaxation interval and vice versa were also calculated to check for further cardiac rhythm dysregulation. 

### 2.4. Calculation of Heart Rhythm by Using OpenCV Libraries

The concept of object tracking is observing moving objects in the video by using a frame-by-frame analysis approach [40]. The initial step was importing various libraries that contain classes and functions that can assist in heart rate computation. CV2 library for OpenCV and imutils (a Python package, https://github.com/jrosebr1/imutils accessed on 30 July 2021) [41] supported in resizing, shaping, translating, rotating, skeletonizing, displaying matplotlib images, sorting contours, and detecting objects. The other libraries like math (https://docs.python.org/3/library/math.html accessed on 30 July 2021) [42], NumPy (https://numpy.org accessed on 30 July 2021) [43], pandas (https://pandas.pydata.org accessed on 30 july 2021) [44], and matplotlib (https://matplotlib.org accessed on 30 july 2021) [45] supported in solving mathematical problems, matrices, data analysis, and plotting graphs, respectively. SciPy (https://www.scipy.org accessed on 30 July 2021), [46] is a library for linearization, optimization, linear algebra, integration, interpolation, special functions, Fast Fourier Transform (FFT) [47], signal and image processing, Ordinary Differential Equation (ODE) solvers, and other tasks common in science and engineering [48]. Argparse library (https://docs.python.org/3/library/argparse.html accessed on 30 July 2021) [49] was used to instantiate command lines that help the code with much information and direction before starting. The overall schema of the study is shown in Figure 1.

### 2.5. Proposed Framework

The initiation of the algorithm will create a loop, loading the video or the file path, resizing the frames which will be processed. Next, bounding boxes are applied to the atrium and ventricle area of the zebrafish heart in the video. Region of interest (ROI) was defined in the raw video, which is well defined and described in the Viola–Jones algorithm [50] in the study of face detection manually [51]. In this study, we used to make a bounding box that contains pixels in the background in greyscale for both ROIs (atrium and ventricle) with no fixed ratio in order to define more precise ROIs subdivisions [52]. The quality of video and the noises when transferring RGB signal to greyscale were optimized by a stringent approach that has a large array of data within ROIs depending upon the quality of signals from video sequences. A subset of ROIs was extracted by using OpenCV, Scipy, and Numpy modules. Tracking was performed directly in determining the ROI of each frame, which was challenging to detect peaks due to the noise in the video. Spline interpolation was then used to get smoothen peaks to overcome the noisy signal problem [53]. To consider of the atrium and ventricle chamber visibility, it was difficult to extract data from the ROIs due to computational complexity. Therefore, the moving subject of the video within ROI was checked repeatedly within 10 s intervals in the 60 fps video dataset. The Mbox (https://docs.python.org/3/library/mailbox.html accessed on 2 August 2021 was used to create a message box, which prompts the user to select the respective ROI. Mbox from the ctype library is C compatible and could access C file using DLL (https://www.dll-files.com accessed on 2 August 2021 or convert the C libraries (https://docs.python.org/3/extending/extending.html accessed on 2 August 2021 to Python. 

In order to get a precise heartbeat value, we used the function ‘def round_half_up’ hat round off float’s values to the nearest integers and is available in the math library. The normalized array function was available in the NumPy library for the standard deviation and mean evaluation. The ‘get similarity’ function was used to compare each data frame by applying a cross-correlation technique. The correlation technique was used for the observational relationship of the different sets in the data, based on their characteristics [54]. Various correlations could also be used to do quantitative correlation. In this study, we used a cross-correlation algorithm to measure the linear relationship in selected ROIs using Python’s multiprocessing [55]. Cross-correlation is applied to measure the linear relationship of a given input video. Cross-correlation gives a ratio of the covariance of two features, and it only took the real value of the range as negative one to positive one (−1 to +1). It was based on a positive linear relationship between the two variables. In order to find the overlap signal values, the Find Peak function was applied first by converting the values of the frame into matrices, which are known as signals (value) in the NumPy library. 

The spline interpolation [53] was applied by using the SciPy library to regulate the video frames with gradual transitions and generate raw signals. The purpose of using spline interpolation is to establish special effects in cardiac rhythm detection. The consecutive sequences of video frames continue to categorize the pixels into good- or low-quality videos. Spline interpolation helps to control fade, dissolved, and wipe features in typical video style [56]. In addition, to overcome brightness in the low-quality video, we dissolved the fade scenes by assuming transitions to strict linear symmetric fashion. The two-blending motion in the video as atrium and ventricle were stopped before selecting the ROI, and then the total number of fps being extracted from the spline interpolation method. Later, data were extracted to measure the heart rate for each cardiac chamber. Finally, we conducted side-by-side data validation by comparing cardiac rhythm obtained from ImageJ and OpenCV for a particular video. 

### 2.6. Extraction of Video Signals from Bounding Box

The convolution process is achieved by taking two matrices of the same dimension and multiplying the element with the element in the selected ROI. Paul Viola and Michael Jones introduced the bounding box concept, which was firstly developed for a frontal face detection system in 2001 [50]. In this study, we applied a comparable application of the bounding box for selecting a specific position of the atrium and ventricle in zebrafish based on expert opinion. The ROI is a subset of selected frame [57], and it has several types as Polygon, Polyline, Point Rectangle, and Ellipse. A rectangular bounding box was applied for the ROI selection as OpenCV has a powerful point-to-point boundary array metrics feature that helps to extract the frame binarization. It is necessary to select the ROI position accurately as it may affect the results due to a false position in the sliding motion of the video. 

Video signals were subjected to NumPy by extending in a matrix, which helps recover and predict signals containing initial unknown information. Raw signal extraction was done by the position of the ROI where video processed frame by frame and greyscale values. Traditionally, color channels i € t [58], were extracted in series as t(i) for the time interval in seconds. We processes the video from color to greyscale image so that pixels can be computed more conveniently as dark and light in video frames. Pixel value computed using correlation algorithm for the highest peak atrium that represented as atrium is pushing out blood to the ventricle. Similarly, the lowest peak showed when the atrium is empty and simultaneously the values computed for ventricle heart chamber following the same principle. The computed pixel values in greyscale were different from the rest of the values that are out of the selected ROIs. A difference of −1 to 1 was applied to calculate the rhythm in each cycle when blood-filled and empty in each heart chamber.

The position of selected ROIs in the video was based on the expert opinion for atrium and ventricle; thus, the quality of video matters a lot in selecting ROIs. Carefully, ROIs selection was performed and extracted raw signals as high peaks atrium and low peaks atrium, high peaks ventricle and low peaks ventricle. The data of peaks was used and saved in csv format for the analysis and validation of results. The heartbeat in a different frames, 30 fps and 60 fps video, was further convolved to get a matrix of two input ROIs, then generated an array of data. The data was then extracted to form signals that were used to imitate the heartbeats of zebrafish. The signal was then plotted based on the data showing a number of frames extracted to the csv file and cardiac rhythm calculated later on as atrium and ventricle heart rate.

### 2.7. Heart Rate and Cardiac Rhythm Measurement

The heart rate measurement was performed by using the OpenCV library with optimization in selected ROI for the heartbeat. The frequency of the deflated image was computed by multiplying the fps value to time in seconds (60) to get bpm. In order to maintain the differential between the next images, we fetched the average values and calculated each interval by making an array of differentials in real-time with peaks. The conventional methods showed limitations to detect a heartbeat from video. Some of the methods, such as a high-speed confocal microscope, are used to detect the faster movement of heartbeat, which could be expensive due to instrumental settings. Our novel method requires mp4 (MPEG-4) files that are widely used to store audio format videos and does not need to be converted to the uncompressed AVI format [26,59].

Our program benefits from facilitating AI-based collection from selected ROI in either ventricle or atrium positions. Next, video information was automatically detected via the OpenCV tracking algorithm. If the ROI was away from the selecting cycle, the signal detection time interval was from 10 s was recorded on 200 fps for 10 s and 60 fps for 10 s, respectively. The video frame rate was reduced to 30 fps to make it compatible with the analysis. By dividing 1 over 200, we get the rate of 0.005, while processing for 60 fps, we must divide 1 over 60, 0.016 value assigned to satisfy the fps range. The total number of frames in peaks obtained 200 × 10 = 2000 (where 200 fps recording for 10 s) and 60 × 10 = 600 (where 60 fps recording performed for 10 s). It is independent of the duration of the video that plays in the system and the number of frames computed above 2000 and less than 10, which were excluded for accurate peaks detection followed by the principle [26]. The ratio was compared between each video dataset as 30 fps, 60 fps, and 200 fps to get accurate heartbeat values the message box created for each video type depending on the scale of the input dataset. For bpm evaluation, we used the following Equation (1):(1)$$\frac{60}{\sum_{i=0}^{i=n-1}0.005*\left(invpeaks_{i+1}-invpeaks_{i}\right)/n}$$
where 60 is the total number of seconds in one minute, *n* = number of frames, and for 200 fps 1 frame = 0.005 and loop of inverse peaks (*invpeaks*) iteration calculated with *n*. Similarly, for 60 fps, 1 frame is equal to 0.016, and by using the above equation, bpm was calculated for a given video dataset. We preliminarily focused on tracking, which can achieve high accuracy with vigorous and illustrative processing. After obtaining raw signals, two signal waveforms obtained from either atrium or ventricle were merged into one panel. Later the original signal waveforms were filtered using bandpass [60] filter to get smoother waveforms and extracted the maxima and minima peak using NumPy and SciPy [61]. Finally, the raw signals over time information can be output in csv format by using the Fast Fourier Transform (FFT) [62] discrete method. Finally, we conducted a mathematical calculation to extract the atrium to atrium (A-A), ventricle to ventricle (V-V), atrium to ventricle (A-V), and ventricle to atrium (V-A) interval endpoints.

### 2.8. Cross-Correlation Algorithm

Heart rate and cardiac rhythm measurement performed using time series cross-correlation standard method [63] followed by the Equation (2) [64]:(2)$$\frac{r=\sum_{1}\left[x\left(i\right)-mx\right)*\left(y\left(i-d\right)-my\right)]}{\sqrt{\sum_{1}(x\left(i\right)-mx^{2}}\sqrt{\sum_{1}{\left(y\left(i-d\right)-my\right)}^{2}}}$$

Considering the series, it has been covered that *x(i)* and *y(i)* are the two-time series and *i* = 0, 1, 2, …, N-1. The cross-correlation *r* and delay *d* expressed. Where *mx* and *my* are defined as means of corresponding series and all the computed differences in frame-by-frame duration *d* = 0, 1, 2, …, N-1, the results showed the twice length from the original series. 

The basic principle implemented in the study to recognize the deflated frames from the ROI and average heart rate during the time of video comprised upon the frequency with the physiology of zebrafish cardiac rhythm. In order to resolve the delays in time as the video being initialized to select ROI, the index in series is considered to be less than 0 or greater than or equal to the number of points wrapped in the motion of frames. The signal processing practices depicted [65] ignoring the points or assuming the series as *x* and *y* is zero. We used the following Equation (3) [64] to compute the index points to satisfy the correlation equation:(3)$$\;\frac{r\left(d\right)=\sum_{1}\left[x\left(i\right)-mx\right)*\left(y\left(i-d\right)-my\right)]}{\sqrt{\sum_{1}(x\left(i\right)-mx^{2}}\sqrt{\sum_{1}{\left(y\left(i-d\right)-my\right)}^{2}}}$$
where *d* represents the range of delays, the length of cross-correlation series was considered less than N (number) when short delays must compute in frames. The −1 to +1 is the typical range that shows maximum correlation and 0 indices as no correlation in each frame. The correlation used to detect the frame recording based on the number of fps as data included in this study have 200 fps and 60 fps. Thus, the duration of high-speed recording video was 10 s. To ensure the pixel-to-pixel ratio in each grayscale frame of selected ROI, the total duration of video was multiplied by 10 to handle the motion of video frames. As frame defined on two-dimensional function *f*(*x*,*y*) as *x* and *y* are the spatial plane coordinates, and the amplitude at respective point shows the grey intensity. 

### 2.9. Smoothing of Raw Data Using Spline Interpolation

We used the spline interpolation method [66] to conduct peak smoothening due to the low error rate between the couple data points. Cubic spline approximate data was defined as piecewise polynomials so that low-order polynomials can be joined together. *Knots* were defined as points where spline must pass, and the total number of points on the curve exist in knots and are determined by the polynomial [67]. The cubic splines are generally preferred due to their low error rate compared to other polynomial splines, which can effectively minimize the root-mean-squared-error (RMSE) between polynomial and data points [58]. We used univariate spline interpolators to smooth the peaks from raw data. While the SciPy library was based on numerical routines in Python programming, it enables the users to perform modeling and solving scientific problems. It includes algorithms for optimization, interpolation, algebraic equations, etc. [68]. SciPy is built with NumPy [69] that contains array data structures with related fast numerical routines. The UnivariateSpline module is available with API for best interaction with SciPy and FITPACK [70]. The shape was confined in a layer of polynomial classes, and the CubicSpline module was constructed to regulate the interpolator in preserved shape [46]. Later, time interval refers to a simple function represented in the graph (Figure 2C) as the number of frames extracted (Figure 2D) on which video was being recorded [71]. Finally, the OpenCV script for zebrafish cardiac rhythm measurement can be found in Appendix A.

## 3. Results

### 3.1. Overview of Heart Rate and Rhythm Analysis Pipeline by OpenCV

In previous studies, our team has developed several simple and cost-effective ImageJ-based methods to conduct cardiac rhythm detection in zebrafish and Daphnia [21,26,72]. In order to effectively reduce the relative complexity for manual or semi-automated operation, in this study, we aimed to develop an automated tool for cardiac rhythm detection in zebrafish by using OpenCV. As shown in Figure 1, we first recorded high-resolution video with a high frame rate from high-speed CCD mounted on an inverted microscope. The traditional approach in image data recording adopts 24-bit RGB three-color space. In order to get a better image resolution (signal-to-noise ratio), the heart area was recorded in grayscale through the frame recognition method in this study. Later, videos in mp4 format were opened in OpenCV software which is implemented in a Python environment. ROI positions at either atrium or ventricle were selected. The recognition algorithm was applied based on a typical rectangular zone representing the heart chamber feature when the heart rate was detected. The detailed standard operational protocol for the tool was available in Appendix A.

In order to verify the accuracy of heart rate detection, the vertical axis in the frames of ROI indicated the beat per minute with the selected position. The video recording was recorded for 10 s at a frame rate of 200 fps. Later, the original videos were rendered to frame rate at 30 fps to make a slow-motion video with a 6.6-fold slowdown effect. Each interval value in the beat cycle was then multiplied with the 6.6 due to the 10 s interval conversion to 66 s to satisfy the original 200 fps rate. In order to calculate atrium and ventricle heart rate multiprocessing in Python [73], we applied a separate ROI option to select the atrium and ventricle parts in the video (Figure 2A). This setting allows us to simultaneously extract two signals from both the atrium and ventricle heartbeat (Figure 2A). 

The Channel and Spatial Reliability algorithm has been considered one of the most accurate in tracking objects globally [32]. OpenCV 3.4.2 version was installed through Python [74]. Since high-speed CCD was used for image capture, we could obtain cardiac rhythm signals with a high sampling rate over time (Figure 2B). Later, the original waveform signals were filtered, and curve smoothening were performed to obtain maxima and minima peak search (Figure 2C). The periodic motion of the heartbeat performs a function, and because of this heart rate signal reaches the maximum in power. Finally, the raw cardiac rhythm data can be output in csv format for information extraction (Figure 2D). The results performing spline interpolation were reproducible (Figure 2C), and it was readily implemented to apply in UnivariateSpline function in Python. 

### 3.2. Cardiac Rhythm Comparison with ImageJ Method in Control Zebrafish Embryos

The basic principle for heart rate and rhythm measurement by OpenCV is different from ImageJ dynamic pixel change or kymograph-based method. In order to know whether data consistency was obtained from OpenCV and ImageJ methods, we compared heart rate from either ventricle or atrium in control zebrafish embryos aged at 72 hpf by using both OpenCV (this study) and ImageJ TSA methods [26]. In Figure 3A, we define some important endpoints to evaluate cardiac rhythm regularity, including atrium to atrium (A-A), ventricle to ventricle (V-V), atrium to ventricle (A-V), and ventricle to atrium (V-A) intervals. Results showed that heart rate was measured by using either OpenCV or ImageJ method displayed a similar level with no significant difference (Figure 3B). The average heart rate in the ventricle was 117 ± 30 for OpenCV and 117 ± 30 for ImageJ (*p* = 0.299); heart rate in the atrium was 118 ± 30 for OpenCV and 116 ± 30 for ImageJ (*p* = 0.316). By Pearson correlation test, the heart rate obtained by both methods displayed a high positive correlation in either ventricle (r = 0.945) or atrium (r = 0.941). Taken together, those pieces of evidence supported our hypothesis that OpenCV can be used to detect zebrafish larvae heart rate in high precision and unbiased manner. 

Next, we validated data consistency for cardiac rhythm measurement by OpenCV and ImageJ TSA methods [26]. For A-A (*p* = 0.891) and V-V (*p* = 0.947) intervals, there are no significant differences be detected between OpenCV and ImageJ TSA methods (Figure 3C). A-V interval was 0.119 ± 30 for OpenCV and 0.152 ± 30 for ImageJ (*p* = 0.324). V-A interval was 0.386 ± 30 for OpenCV and 0.361 ± 30 for ImageJ (*p* = 0.445). For the A-A interval (Figure 3E) and V-V interval (Figure 3F) regularity test, we found OpenCV can detect more uniform heartbeat regularity by showing significantly less sd1 and sd2 values than those conducted by the ImageJ TSA method. Taken together, this evidence supported our hypothesis that OpenCV can be used to detect zebrafish cardiac rhythm in high precision and unbiased manner. The comparison results also support that the current OpenCV approach has consistent results with the ImageJ TSA method.

### 3.3. Cardiac Rhythm Comparison between OpenCV and ImageJ Methods with Different ROI Selected Positions

Next, we tested the potential effects of ROI position in both atrium and ventricle for cardiac rhythm measurement. Three different locations within the heart chamber, bottom, middle, and top positions, were selected for comparison (Figure 4C). Afterward, we tested the difference of ROI location selection by comparing the atrium-ventricle (A-V) and ventricle to atrium (V-A) intervals from three different ROI location sets, which were middle atrium-middle ventricle, bottom atrium-top ventricle, and top atrium-bottom ventricle. The result showed there were no significant difference in A-V intervals using ImageJ on Middle (0.1666 ± 0.1387), Bottom A-Top V (0.1250 ± 0.07164), and Top A-Bottom V (0.1756 ± 0.1161) and OpenCV on Middle (0.1614 ± 0.1369), Bottom A-Top V (0.1453 ± 0.1209), and Top A-Bottom V (0.1192 ± 0.06383) (Figure 4A). The result on V-A intervals also showed no significant difference between ImageJ on Middle (0.3346 ± 0.1359), Bottom A-Top V (0.3788 ± 0.0644), and Top A-Bottom V (0.3284 ± 0.1215) and OpenCV on Middle (0.3402 ± 0.1386), Bottom A-Top V (0.3590 ± 0.1183), and Top A-Bottom V (0.3854 ± 0.06457) (Figure 4B). Therefore, OpenCV method established in this study is insensitive to the ROI position and can get consistent results comparable to the ImageJ TSA method.

### 3.4. Use of the OpenCV Method to Detect Heart Rate Alterations after IBMX Treatment

Next, we examined the performance of OpenCV to measure tachycardia (heartbeat faster) events. IBMX (3-Isobutyl-1-methylxanthine) was used to induce tachycardia, according to our previous published literature reported by Santoso et al. [75]. For IBMX, we found a significant increase in heart rate and decrease in heartbeat interval in both atrium and ventricle chambers compared to the control fish (Figure 5A,B). The decrease in heartbeat intervals (Figure 5C,D) detected by either OpenCV approach or ImageJ TSA method display no significant difference (for atrium, 0.2432 ± 0.0791 for OpenCV and 0.2436 ± 0.0791 for ImageJ, *p* = 0.3750; and for ventricle, 0.2435 ± 0.0795 for OpenCV and 0.2435 ± 0.794 for ImageJ, *p* = 0.9102). In addition, the decrease in A-V or V-A intervals detected by either OpenCV, or ImageJ TSA method display consistent results (for A-V intervals, 0.1066 ± 0.0337 for OpenCV and 0.0974 ± 0.0256 for ImageJ, *p* = 0.5837; and for V-A intervals, 0.1739 ± 0.0300 for OpenCV and 0.1834 ± 0.0294 for ImageJ, *p* = 0.5677).

To detect cardiac rhythm alteration, we performed the Poincare plot method to measure sd1 and sd2 values (Figure 5E,F). Zebrafish embryos display less cardiac rhythm regularity if they have higher sd1 and sd2 values. For A-A interval, we found sd1 was 0.0221 ± 0.0094 for OpenCV and 0.0421 ± 0.0119 for ImageJ (*p* = 0.0195, *), and sd2 was 0.0188 ± 0.0167 for OpenCV and 0.0294 ± 0.0114 for ImageJ (*p* = 0.0195, *). For V-V interval, we found sd1 was 0.0138 ± 0.0066 for OpenCV and 0.0236 ± 0.0074 for ImageJ (*p* = 0.0020, **), and sd2 was 0.0135 ± 0.0160 for OpenCV and 0.0202 ± 0.0145 for ImageJ (*p* = 0.0039, **). Taking together, these pieces of evidence showed the capability of OpenCV in obtaining tachycardia heartbeat data on IBMX exposed zebrafish, which are comparable to the previously published method [75]. Moreover, by using OpenCV, obtained heart rate variability is more uniform compared to ImageJ.

### 3.5. Use of OpenCV Method to Detect Cardiac Rhythm Dysregulation after Camphor Treatment

Furthermore, we hypothesized that our program was capable of detecting bradycardia (heartbeat slow down) and cardiac dysregulation (arrhythmia). In order to validate our hypothesis, we used a previously published dataset showing heart rate slow down and cardiac rhythm dysregulation after 500 ppm Camphor exposure reported by Du et al. [76]. After 500 ppm Camphor exposure, we found a significant slowdown in the heart rate in both atrium and ventricle chambers compared to control fish (Figure 6A,B). Importantly, the slower heartbeat detected by either OpenCV or ImageJ TSA method display no significant difference for either atrium (88.51 ± 16.88 for OpenCV, and 100.5 ± 29.52 for ImageJ, *p* = 0.2778) or ventricle (84.83 ± 9.505 for OpenCV and 79.44 ± 13.55 for ImageJ, *p* = 0.0571). The heartbeat intervals (Figure 6C,D), on the contrary, are significantly longer than those in control fish after 500 ppm Camphor treatment. The heartbeat interval detected by either OpenCV or ImageJ TSA method showed a comparable result to heart rate. There is no significant difference for A-A intervals which has, 0.6162 ± 0.0789 for OpenCV and 0.6381 ± 0.1604 for ImageJ, *p* = 0.3543; and for V-V intervals, 0.6010 ± 0.0975 for OpenCV and 0.7752 ± 0.1316 for ImageJ, *p* = 0.068. For cardiac rhythm regularity, we performed the Poincare plot method to measure sd1 and sd2 values. Zebrafish embryos display less cardiac rhythm regularity if they have higher sd1 and sd2 values (Figure 6E,F). For A-A interval, we found sd1 was 0.0984 ± 0.0540 for OpenCV and 0.0505 ± 0.0660 for ImageJ (*p* = 0.0371), and sd2 was 0.0980 ± 0.0623 for OpenCV and 0.0338 ± 0.0265 for ImageJ (*p* = 0.0137). For V-V interval, we found sd1 was 0.0864 ± 0.0714 for OpenCV and 0.0205 ± 0.0167 for ImageJ (*p* = 0.0098), and sd2 was 0.0958 ± 0.0738 for OpenCV and 0.0254 ± 0.0159 for ImageJ (*p* = 0.0273). Taken together, these pieces of evidence support our hypothesis that OpenCV can be used to detect zebrafish cardiac rhythm dysregulation in high precision and unbiased manner and suggest that OpenCV is more sensitive than the ImageJ TSA method on detecting cardiac rhythm regularity in zebrafish.

### 3.6. Use of OpenCV Method to Detect an Ultrafast Heartbeat in Daphnia Magna

The success of detecting cardiac rhythm in zebrafish by the OpenCV method promotes us to test whether our approach is also functional when applied to ultrafast heartbeat events in water fleas. We analyzed heartbeat data for *Daphnia magna* reported by Santoso et al. [72]. The control peaks for *Daphnia magna* heartbeat analyzed by OpenCV method can be found in Appendix A, with a good signal-to-noise ratio. The heart rate in *Daphnia magna* was temperature-dependent, which was detected 354.2 ± 41.70 bpm by using ImageJ and 354.3 ± 41.94 bpm by using OpenCV at 15 °C (*p* = 0.475, Figure 7A). The heart rate of *Daphnia* increased after being exposed to a higher temperature at 35 °C, reaching 630.8 ± 99.77 bpm using ImageJ and 631.5 ± 99.81 bpm using OpenCV (*p* = 0.241, Figure 7B). Observation on *Daphnia* heartbeat intervals at 15 °C showed average interval of 0.1716 ± 0.01951 s of ImageJ compared to 0.1715 ± 0.01971 s of OpenCV (*p* = 0.504, Figure 7C), while 35 °C showed average interval of 0.09773 ± 0.01769 s of ImageJ compared to 0.09762 ± 0.01766 s of OpenCV (*p* = 0.165, Figure 7D). Both methods showed a comparable interval result with good consistency. Afterward, heartbeat regularity between both methods was compared by using a Poincare plot. sd1 result showed no significant difference between ImageJ and OpenCV method (0.01488 ± 0.00901 s and 0.01018 ± 0.007254 s, *p* = 0.058, respectively) and significant difference on the sd2 result (0.01033 ± 0.01059 s and 0.005825 ± 0.00519 s, respectively) for *Daphnia magna* exposed to 15 °C water temperature. The difference in heartbeat regularity between both methods became more pronounced at 35 °C. ImageJ showed sd1 regularity of 0.007185 ± 0.005131 s, while OpenCV showed a more regular result at 0.004775 ± 0.003995 s. ImageJ showed sd2 regularity of 0.004225 ± 0.002868 s compared to OpenCV of 0.002740 ± 0.001894 s (Figure 7D,E). The sd1 and sd2 values obtained from OpenCV display more uniform than the ImageJ method, which is consistent with previous tests done in zebrafish embryos. 

### 3.7. Use of OpenCV Method to Detect Heartbeat Irregularity in Daphnia Magna

Afterward, we further validated the result obtained in *Daphnia magna* by exposing the model to the pesticide in order to induce heartbeat irregularity. We used a dataset collected from imidacloprid (IMI) exposed *Daphnia magna*, which was reported by Santoso et al. [72]. Compared to the control group (357.7 ± 86.6 bpm), we found IMI exposure at 100 ppb can reduce *Daphnia magna* heartbeat to a level of 315.1 ± 116.7 by ImageJ and 309.4 ± 124.2 bpm by OpenCV, which showed no significant difference between both methods (*p* = 0.855, Figure 8A). For heart rate interval, compared to the control group (0.1779 ± 0.04596 s), IMI exposure at 100 ppb can prolong Daphnia heartbeat interval to a level of 0.2447 ± 0.1674 by ImageJ and 0.2631 ± 0.1919 by OpenCV with no significant difference between both methods (*p* = 0.7609, Figure 8B). Next, we used sd1 and sd2 values from the Poincare plot to conduct the heartbeat regularity test. Results show IMI exposure can significantly elevate sd1 and sd2 levels in *Daphnia magna*, and OpenCV can generate more uniform results (sd1 = 0.03349 ± 0.04744, sd2 = 0.04110 ± 0.06569) than ImageJ (sd1 = 0.07052 ± 0.1164, sd2 = 0.07349 ± 0.1124) for heartbeat regularity measurement (Figure 8C–E). 

## 4. Discussion

The most important findings for this study are that we adopted a Computer Vision-based OpenCV approach for conducting cardiac rhythm detection in zebrafish for the first time. Compared to previous manual or semi-automated ImageJ-based methods introduced by Sampurna et al. [26] and Kurnia et al. [21], this new OpenCV-based method has several advantages of high accuracy, easy operation, and automated data process pipeline. Previous literature has conducted a good comparison between OpenCV and ImageJ methods [77,78]. ImageJ is a Java-based tool originally developed by the National Institute of Health (NIH) to analyze images or videos and successfully contribute to many fields [79]. The major reasons for its success are an easy-to-use interface, easy-to-get plugins, and a macro system that allows users to capture and get interactions by automatically reproduce workflows. ImageJ has featured mainly focused on image processing. It lacks instrumental tool perspectives to compete with the broad spectrum of advanced positioning in computer vision and machine learning methods. However, there is also a limitation that each plugin requires to get other algorithms either from scratch or must implement the method with connections of other external libraries. The ImageJ-based method that we developed previously for cardiac rhythm measurement is based on manual or semi-automated calculation [21,26]. The user must convert mp4 into AVI uncompressed format and open a number of excel software sheets to compute heart rate, cardiac endpoints (AV-VA), and cardiac rhythm regularity. In the current OpenCV approach, all essential tools have already been installed in the same python platform. Therefore, there is no need to use external software such as VirtualDub, Adobe Premiere CC, and Microsoft excel used in the previous ImageJ-based methods [21,26] to perform cardiac rhythm analysis. 

OpenCV, on the contrary, has numerous libraries established for image processing and object tracking purposes. The architecture is made for systematic analysis in various computational metrics as widths and spacing to work in a bounding box to track the ROI. This concept is common in practice in machine learning approaches. The power of OpenCV relies on image processing, feature detection, object tracking, machine learning, and video analysis. The parameters in tracking as bars, also, subsequently the name with rectangular regions were framed by a various blend of uncommon characters like colons or spots. The library has been developing continuously since its introduction to the world, attempting to use widely in biomedicine and image processing. It provides a wide spectrum to implement code based on the requirement of the user to solve the problem. OpenCV displays more flexibility for accepting multiple video input formats than ImageJ and can skip the tedious and time-consuming video format conversion step. Video frame rates set at either 30, 60, or 200 fps are acceptable and can be automatically handled with each input video recording frame size. 

ImageJ is used to detect pixels in an image, while the OpenCV-based tool tends to perform computation in ROIs within video files as the video has much more efficiency to get good results compared to a single image. Python multiprocessing tool allows users to simultaneously conduct atrium and ventricle heart rate measurements using multiple ROI selection functions. In contrast, ImageJ only allows selecting a single ROI each time for either atrium or ventricle heart chambers selection and extracting the data from peaks for only one ROI at each time. Finally, we also provide standard operating protocols (SOP) as Appendix A that allow users to conduct cardiac rhythm measurements without any training on Python scripting. The user can operate our developed OpenCV tool in a simple way just conduct copy and paste commands in the Python operation environment.

In order to validate our result, we compared the result of our OpenCV method to a previously published method using a time series analyzer in ImageJ [26]. First, we compared three different ROI locations using both methods. This comparison resulted in no significant difference between different ROIs. The result means both methods are not affected by different area selection, which means they will reduce the possible error during the ROI selection process. We also validated that OpenCV was able to detect tachycardia, bradycardia as well as arrhythmia events for zebrafish larvae with high precision comparable to ImageJ TSA method. Therefore, this new, innovated OpenCV tool can provide the research community with a convenient and low-cost tool to conduct cardiotoxicity or cardio-pharmacological assay. Table 1 Presented recent studies on automated and manual methods to measure heart rate and cardiac rhythm in zebrafish.

## 5. Potential Limitations and Future Work

This study successfully developed an OpenCV-based cardiac rhythm detection tool for zebrafish and other aquatic invertebrates (like water flea). The good utilization of this OpenCV tool has been extensively validated by a side-by-side comparison to previously ImageJ-based methods [26]. The proper ROI selection has been reported to play a crucial role in a kymography-based method for cardiac rhythm detection [21]. ROIs defined in the current work are following the mounting so broadly to target the atrium and ventricle to avoid the motion sliding in video frames. Each video frame was read to cover ideal spatial differences for handling the enormous ROI selections, and calculations are wise as they can be used to plot cardiac rhythm data. The key computational significances are dynamic, measuring beat per minute containing deflated accessible data with spline interpolation plots. The current OpenCV-based method required a minimal video length for at least 10 s but no limitation for longer videos. For ultrafast heartbeat detection in *Daphnia magna*, we found that separate scales are required based on the different ratios for reading frames. This is because the ratio of scale setting for zebrafish does not work for *Daphnia magna* cardiac rhythm analysis. 

Finally, the quality of input video sequences plays an important role in rhythm detection. Poor recording practices with low image contrast will reduce the accuracy of cardiac rhythm detection. In our OpenCV tool, we found although this method is not sensitive to ROI position selection in the heart chamber; however, the user still needs to conduct ROI manual selection step. Since the ROI selection in each video was different and should be manually selected, the current version does not support batch conduction. The future study by machine learning to conduct automated ROI selection might overcome this limitation [81]. Furthermore, in the future, the addition of Graphical User Interface (GUI) could benefit a lot of people who use this tool, as the addition of GUI will improve the usability of the tool, especially for researchers without a computer science background.

## 6. Conclusions

In this study, we developed an OpenCV-based tool that can perform automated cardiac rhythm measurement of both atrium and ventricle chambers in zebrafish larvae and the ultrafast heartbeat in *Daphnia magna* video datasets. The program can examine the ROI with spatial deflated frame scale and provide results in a flexible manner to analyze further for cardiac endpoints as heartbeat regularity. This tool also provides one more scope to get enough information to perform multiple analyses such as non-contact monitoring, image processing, and ambulatory cardiac activity with attractive potential for pharmacological and toxicological screening application. 

## Figures and Tables

**Figure 1 biomolecules-11-01476-f001:**
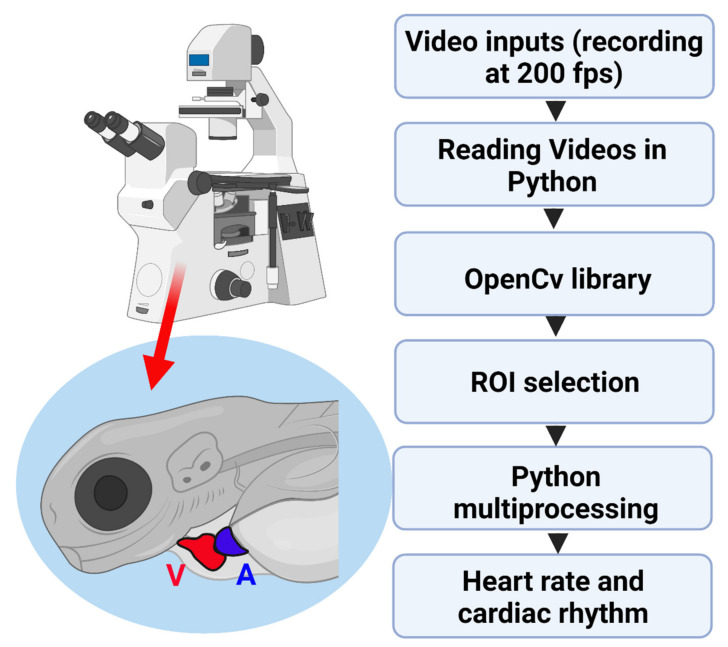
Schematic diagram showing the instrument setting and analysis pipeline for OpenCV-based cardiac rhythm detection in this study. A: atrium, V: ventricle.

**Figure 2 biomolecules-11-01476-f002:**
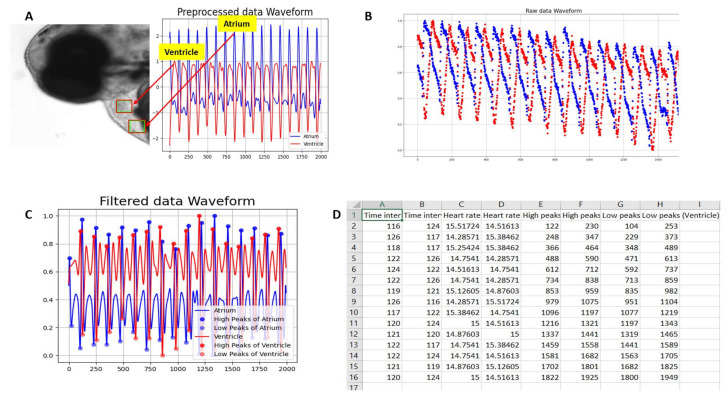
(**A**) ROI selection for ventricle and atrium in zebrafish heart video. The amplitude on the y-axis and the number of frames on the x-axis against the ROI in the selected video were presented. The division algorithm was to find the peaks based on each frequency of selected frames. (**B**) Raw heartbeat signals obtained from OpenCV. Blue peaks showing atrium heart rate, and red peaks presented ventricle heart rate. The X-axis contains the number of frames, and the y-axis shown amplitude based on pixels intensity. Overlay graph having atrium and ventricle beat per minute results. (**C**) The filtered heartbeat waveforms for both the atrium (blue color) and ventricle (red color). (**D**) Output data format in excel showing some important measurements.

**Figure 3 biomolecules-11-01476-f003:**
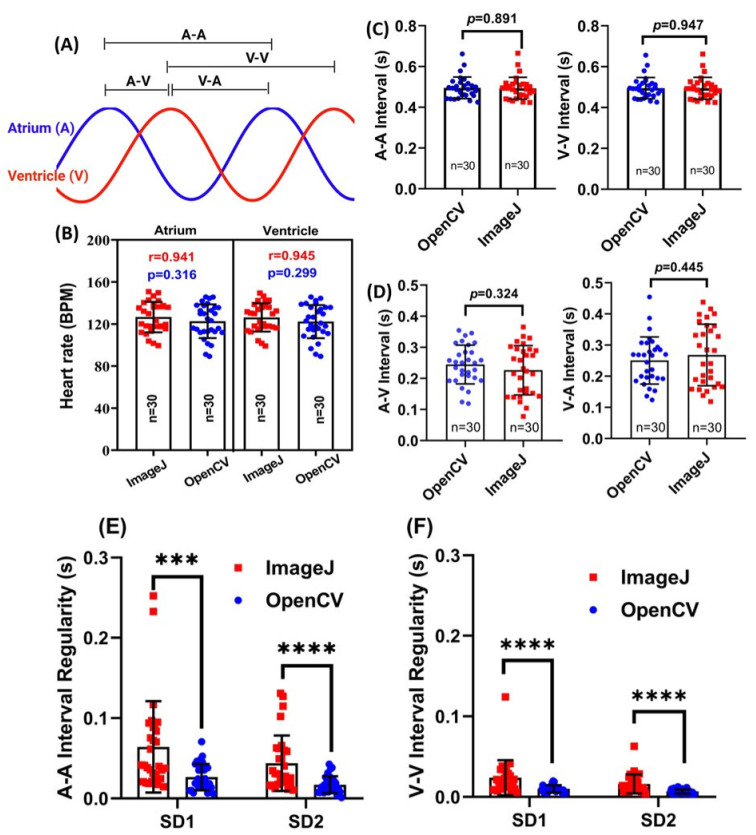
Comparison of heart rate and cardiac rhythm measurement in control zebrafish embryos aged at 72 hpf using either OpenCV or ImageJ methods. (**A**) Schematic diagram showing the basic definition for atrium and ventricle signals. (**B**) Comparison of heart rate obtained from either atrium (left panel) or ventricle (right panel) chambers using either ImageJ or OpenCV methods. (**C**) Comparison of the atrium to atrium (A-A) and ventricle to ventricle (V-V) intervals using either ImageJ or OpenCV methods. (**D**) Comparison of atrium-ventricle (A-V) and ventricle to atrium (V-A) intervals using either ImageJ or OpenCV methods. Comparison of atrium-atrium (A-A) (**E**) and ventricle-ventricle (V-V) (**F**) interval regularity using either ImageJ or OpenCV methods. Data was processed using unpaired *t*-test (**B**–**D**) as it follows the normal distribution. Pearson correlation (**B**) and Wilcoxon test (**E**,**F**) were conducted because the data does not obey normal distribution (*** *p* < 0.001, **** *p* < 0.0001).

**Figure 4 biomolecules-11-01476-f004:**
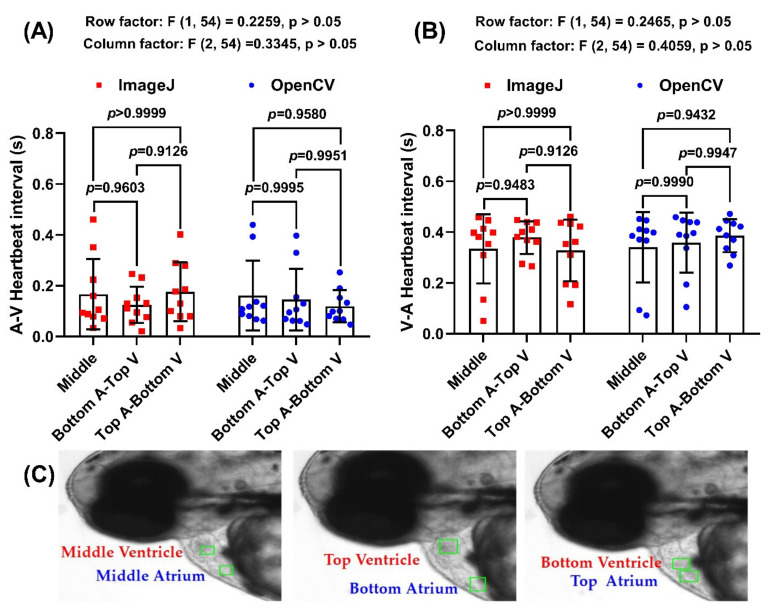
Comparison of Atrium-Ventricle (**A**) and Ventricle-Atrium (**B**) heartbeat interval using ImageJ and OpenCV method. (**C**) Schematic diagram showing the position for ROI selection. Data was processed using Two-Way ANOVA with Tukey comparison method and presented as Mean ± SEM; *n* = 10.

**Figure 5 biomolecules-11-01476-f005:**
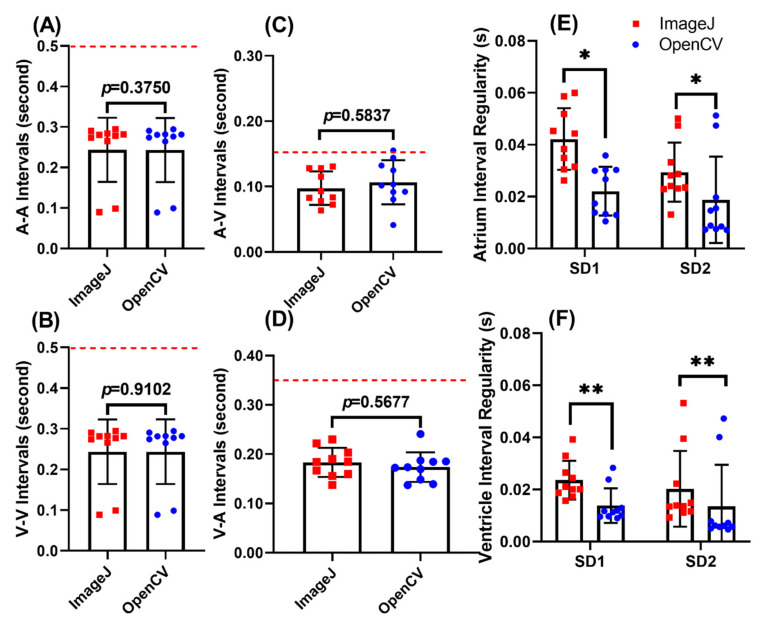
Comparison of heart rate regularity and cardiac rhythm measurement in IBMX treated zebrafish embryos aged at 72 hpf using either OpenCV (red color) or ImageJ (blue color) methods. Comparison of atrium to atrium (A-A) (**A**) and ventricle to ventricle (V-V) (**B**) intervals using either ImageJ or OpenCV methods. Comparison of atrium-ventricle (A-V) (**C**) and ventricle to atrium (V-A) (**D**) intervals using either ImageJ or OpenCV methods. Comparison of heartbeat regularity in either atrium (**E**) or ventricle (**F**) using either ImageJ or OpenCV methods, red dotted line signified the normal level of each parameter obtained from control group. Data was calculated using *t*-test (**A**,**B**) and Wilcoxon test (**C**–**F**), and presented as Mean ± SD, *n* = 10 (* *p* < 0.05; ** *p* < 0.01).

**Figure 6 biomolecules-11-01476-f006:**
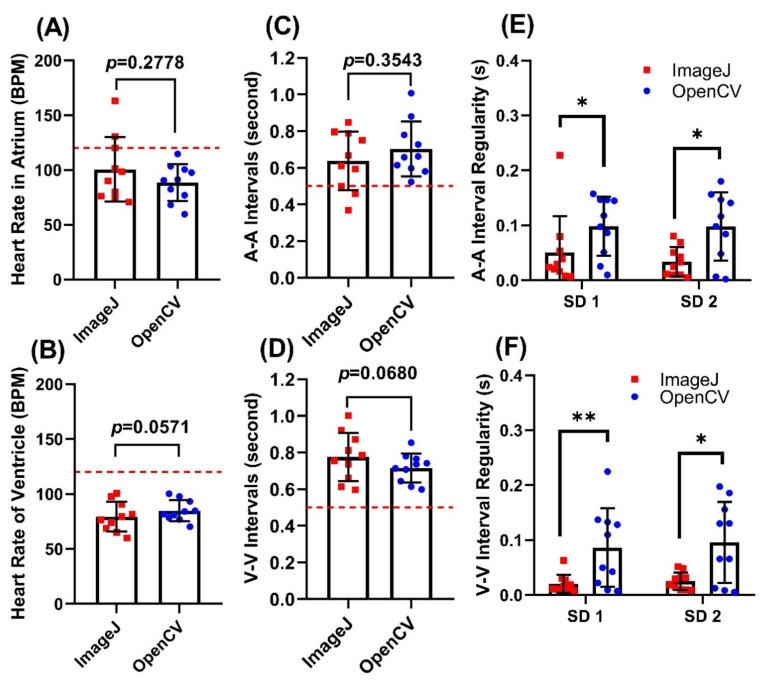
Comparison of heart rate and cardiac rhythm measurement in 500 ppm Camphor-treated zebrafish embryos aged at 72 hpf using either OpenCV or ImageJ methods. Comparison of heart rate obtained from either atrium (**A**) or ventricle (**B**) chambers using ImageJ TSA and OpenCV methods. Comparison of atrium to atrium (A-A) (**C**) and ventricle to ventricle (V-V) (**D**) intervals using ImageJ and OpenCV methods. Comparison of heartbeat regularity in either atrium (**E**) or ventricle (**F**) using either ImageJ or OpenCV methods, red dotted line signified normal value of each parameter obtained from control group. Data was calculated using *t*-test (**A**,**B**) and Wilcoxon test (**C**–**F**), and presented as Mean ± SD, *n* = 10 (* *p* < 0.05; ** *p* < 0.01).

**Figure 7 biomolecules-11-01476-f007:**
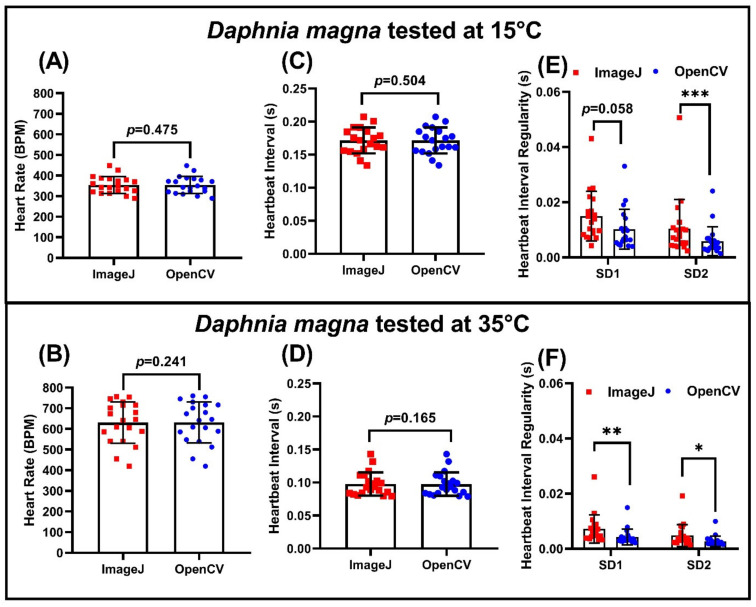
OpenCV can be used to detect an ultrafast heartbeat in *Daphnia magna.* Comparison of heart rate obtained from at either 15 °C (**A**) or 35 °C (**B**) using ImageJ TSA and OpenCV methods. Comparison of heartbeat intervals at either 15 °C (**C**) or 35 °C (**D**) using ImageJ and OpenCV methods. Comparison of heartbeat regularity at either 15 °C (**E**) or 35 °C (**F**) using ImageJ TSA and OpenCV methods. Data was calculated using Wilcoxon test and presented as Mean ± SD, *n* = 20 (* *p* < 0.05; ** *p* < 0.01; *** *p* < 0.001).

**Figure 8 biomolecules-11-01476-f008:**
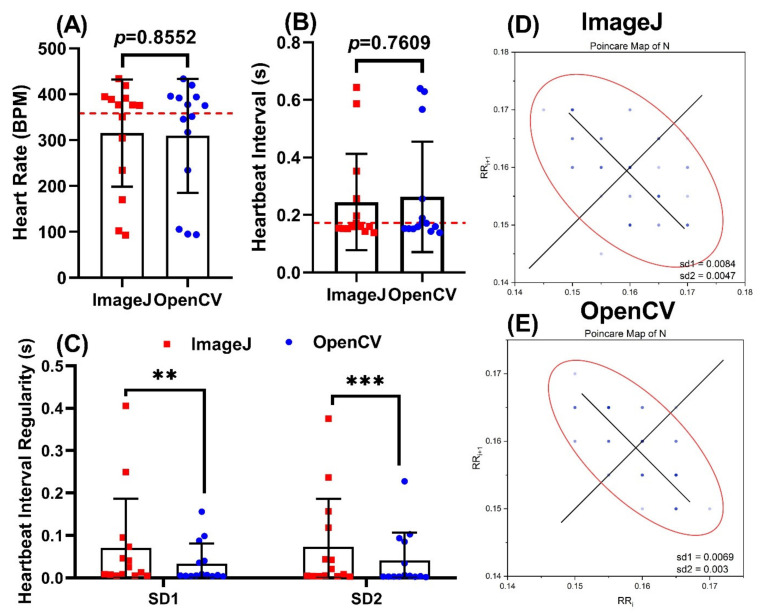
OpenCV can be used to detect heartbeat irregularity in *Daphnia magna.* Comparison of heart rate (**A**) and heartbeat interval (**B**) obtained using ImageJ TSA and OpenCV methods. Comparison of heartbeat regularity using ImageJ and OpenCV methods (**C**), red dotted line signifies normal value of each parameter obtained from control group. Poincare plots were used to measure heartbeat regularity for either ImageJ (**D**) or OpenCV (**E**) methods. Data was calculated using Wilcoxon test and presented as Mean ± SD, *n* = 14 (** *p* < 0.01; *** *p* < 0.001).

**Table 1 biomolecules-11-01476-t001:** Comparison of OpenCV and other methods for heart rate and cardiac rhythm measurement in zebrafish.

Software	Video Format	Environment/Platform	Selecting ROI	Automated Operation?	Endpoints Can be Measured	References
OpenCV	MP4	Python	Video	Yes	Heart rate, atrium-atrium, ventricle-ventricle, atrium-ventricle, and ventricle-atrium intervals, heart rate variability	This study
ImageJ TSA	AVI	ImageJ	Image	No	Heart rate, atrium-atrium, ventricle-ventricle, atrium-ventricle, and ventricle-atrium intervals, heart rate variability	[26,80]
Kymograph	AVI	ImageJ	Image	No	Heart rate, atrium-atrium, ventricle-ventricle, atrium-ventricle, and ventricle-atrium intervals, heart rate variability	[21]
ZACAF (Zebrafish Automatic Cardiovascular Assessment Framework)	Not mentioned	Python (U-net deep learning model)	Video, Selected by deep learning tool	Yes	Heart rate only	[81]

## Data Availability

The data presented in this study are available on request from the corresponding author.

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
