# Peer review of "An OpenCV-Based Approach for Automated Cardiac Rhythm Measurement in Zebrafish from Video Datasets"

_biomolecules, 2021, doi:10.3390/biom11101476_

Round 1

Reviewer 1 Report

The work aims to provide enough evidence that OpenCV analysis of Zebrafish cardiac rhythm and its variation is as good as ImageJ based previously validated technique. The authors succeeded on showing the suitability of OpenCV based approach through several measurement done in parallel with the current standard. The methodology of the work is well-structured, and of good quality.

Some suggestions to the authors:

line 51 should be rephrased since it is not clear, 
lines 51-53, the sentence seems to be too long,
line 394 where it reads V-V interval (Figure 3E) should be V-V interval  (Figure 3F),
I would also recommend the authors to review test format (extra spaces found frequently throughout the text and different front type and size).

Since the OpenCV is described as "the user can operate our developed OpenCV tool in a simple way just conduct copy and paste commands in the Python operation environment." I would like to see an image of the interface of this tool and how would it look to the user (maybe as a supplementary material).

I also suggest the authors to write a small description on how to open the video available as Video S1, since it might not be obvious for people that do not work in Python.   

Author Response

Comments and Suggestions for Authors

The work aims to provide enough evidence that OpenCV analysis of Zebrafish cardiac rhythm and its variation is as good as ImageJ based previously validated technique. The authors succeeded on showing the suitability of OpenCV based approach through several measurement done in parallel with the current standard. The methodology of the work is well-structured, and of good quality.

Some suggestions to the authors:

line 51 should be rephrased since it is not clear 

Thank you for the suggestion. The author also agree that he sentences in line 51 need to be rephrase for better understanding of the authors intention. Thus, the manuscript has been revised according to the reviewer suggestion.

 lines 51-53, the sentence seems to be too long,

Thank you for the suggestion. The authors also agree that the sentence in the pointed part was too long. Thus, the sentences has been broken up into several sentences to improve the reader comprehension. In the revised manuscript.

line 394 where it reads V-V interval (Figure 3E) should be V-V interval  (Figure 3F), I would also recommend the authors to review text format (extra spaces found frequently throughout the text and different front type and size).

Thank you for thorough review. The authors already made some change in the pointed part and more detail check has been done to the other part regarding the text format in the revised manuscript.

Since the OpenCV is described as "the user can operate our developed OpenCV tool in a simple way just conduct copy and paste commands in the Python operation environment." I would like to see an image of the interface of this tool and how would it look to the user (maybe as a supplementary material).

Thank you for the suggestion. At the time of the paper writing, the tools did not have any graphic user interface (GUI) because the tools work by copy the command line into the Python operation environment as it show one Video S1. However, the authors agree that creating GUI will benefit a lot of people who use this method especially the one without any computer science background. Thus, this matter has been added in the revised manuscript at future work section.

I also suggest the authors to write a small description on how to open the video available as Video S1, since it might not be obvious for people that do not work in Python.   

Thanks for valuable suggestions. The video tutorial that show the standard operation protocol to use the tool has been uploaded to YouTube (recognized social media platform by Google) and the link to open the video was added in supplementary SOP (File S2).

Reviewer 2 Report

During the last few years the importance of zebrafish as a model organism for the study of cardiovascular diseases has increased exponentially. For this reason the manuscript of Ali Farhan and colleagues describing the development of an OpenCV-based tool that can perform automated cardiac rhythm measurement of both atrium and ventricle chambers in zebrafish larvae is of interest. Besides zebrafish, the authors show that this tool can be also successfully used for cardiac rhythm analysis in other organisms with extremely fast heartbeat.

Although the language has some problems and needs to be revised, the manuscript is clear, the experiments well conducted and the comparison with a widely used method satisfactory. The advantages of the new instrument are also clearly presented.

Daphnia magna is a good model for testing the reliability and reproducibility of this OpenCV method even with organisms showing an extremely high heart rate. However, this is not a widely used model organism so the test to verify the ability of OpenCV method to detect heartbeat irregularity should be carried out in zebrafish embryos for which several pro-arrhythmic drugs are known.

Author Response

Comments and Suggestions for Authors

During the last few years the importance of zebrafish as a model organism for the study of cardiovascular diseases has increased exponentially. For this reason the manuscript of Ali Farhan and colleagues describing the development of an OpenCV-based tool that can perform automated cardiac rhythm measurement of both atrium and ventricle chambers in zebrafish larvae is of interest. Besides zebrafish, the authors show that this tool can be also successfully used for cardiac rhythm analysis in other organisms with extremely fast heartbeat.

Although the language has some problems and needs to be revised, the manuscript is clear, the experiments well conducted and the comparison with a widely used method satisfactory. The advantages of the new instrument are also clearly presented.

Thank you for the valuable comment. More detail language editing has been done in the revised manuscript according to the reviewer suggestion.

Daphnia magna is a good model for testing the reliability and reproducibility of this OpenCV method even with organisms showing an extremely high heart rate. However, this is not a widely used model organism so the test to verify the ability of OpenCV method to detect heartbeat irregularity should be carried out in zebrafish embryos for which several pro-arrhythmic drugs are known.

Thank you for the comment. The test performed in Daphnia was to show the versatility of the tool by showing the capability analyze an extremely high heart rate, while the test to show the capability of the tools to analyze arrhythmic heart rate was done in zebrafish incubated with IBMX and camphor. IBMX was a pro-arrhythmic drug which can increase the heart rate of zebrafish while camphor was reported to be able to cause arrhythmia in zebrafish larvae according to previously published study (Du et al., 2021; Santoso et al., 2019)

Du, Z.-C., Xia, Z.-S., Zhang, M.-Z., Wei, Y.-T., Malhotra, N., Saputra, F., Audira, G., Roldan, M. J. M., Hsiao, C.-D., & Hao, E.-W. (2021). Sub-lethal Camphor Exposure Triggers Oxidative Stress, Cardiotoxicity, and Cardiac Physiology Alterations in Zebrafish Embryos. Cardiovascular Toxicology, 1-13.

Santoso, F., Sampurna, B. P., Lai, Y.-H., Liang, S.-T., Hao, E., Chen, J.-R., & Hsiao, C.-D. (2019). Development of a simple imagej-based method for dynamic blood flow tracking in zebrafish embryos and its application in drug toxicity evaluation. Inventions, 4(4), 65.